# Validation of the Infant and Young Child Development (IYCD) Indicators in Three Countries: Brazil, Malawi and Pakistan

**DOI:** 10.3390/ijerph18116117

**Published:** 2021-06-06

**Authors:** Melissa Gladstone, Gillian Lancaster, Gareth McCray, Vanessa Cavallera, Claudia R. L. Alves, Limbika Maliwichi, Muneera A. Rasheed, Tarun Dua, Magdalena Janus, Patricia Kariger

**Affiliations:** 1International Child Health and Neurodevelopmental Paediatrics, Department of Women and Children’s Health, Institute of Life Course and Medical Sciences, University of Liverpool, Liverpool L12 2AP, UK; 2School of Medicine, Keele University, Keele ST5 5BG, UK; g.lancaster@keele.ac.uk (G.L.); g.mccray@keele.ac.uk (G.M.); 3Brain Health Unit in Department of Mental Health and Substance Use, World Health Organization (WHO), 1202 Geneva, Switzerland; cavallerav@who.int (V.C.); duat@who.int (T.D.); 4Pediatrics Department, Medicine School, Universidade Federal de Minas Gerais (UFMG), Belo Horizonte 30130-100, Brazil; lindgren@medicina.ufmg.br; 5Department of Psychology, University of Malawi, Zomba P.O. Box 280, Malawi; lmaliwichi@cc.ac.mw; 6Centre for International Health, Department of Global Public Health and Primary Care, University of Bergen, 5007 Bergen, Norway; muneera.rasheed@uib.no; 7Offord Centre for Child Studies, Department of Psychiatry and Behavioural Neurosciences, McMaster University, Hamilton, ON L8S 4KI, Canada; janusm@mcmaster.ca; 8Center for Effective Global Action (CEGA), School of Public Health, University of California, Berkeley, CA 94704, USA; patriciakariger@gmail.com

**Keywords:** child development, measurement, indicators, global health, validation, cross-cultural, cross-linguistic

## Abstract

Background: The early childhood years provide an important window of opportunity to build strong foundations for future development. One impediment to global progress is a lack of population-based measurement tools to provide reliable estimates of developmental status. We aimed to field test and validate a newly created tool for this purpose. Methods: We assessed attainment of 121 Infant and Young Child Development (IYCD) items in 269 children aged 0–3 from Pakistan, Malawi and Brazil alongside socioeconomic status (SES), maternal educational, Family Care Indicators and anthropometry. Children born premature, malnourished or with neurodevelopmental problems were excluded. We assessed inter-rater and test-retest reliability as well as understandability of items. Each item was analyzed using logistic regression taking SES, anthropometry, gender and FCI as covariates. Consensus choice of final items depended on developmental trajectory, age of attainment, invariance, reliability and acceptability between countries. Results: The IYCD has 100 developmental items (40 gross/fine motor, 30 expressive/receptive language/cognitive, 20 socio-emotional and 10 behavior). Items were acceptable, performed well in cognitive testing, had good developmental trajectories and high reliability across countries. Development for Age (DAZ) scores showed very good known-groups validity. Conclusions: The IYCD is a simple-to-use caregiver report tool enabling population level assessment of child development for children aged 0–3 years which performs well across three countries on three continents to provide reliable estimates of young children’s developmental status.

## 1. Introduction

Global efforts through the Millennium Development Goals have been very successful in ensuring many more children survive the perinatal period. However, this has not led to sustained thriving of those same children [1] Estimates now indicate that over 250 million children in low- and middle-income countries (LMICs) are at risk of not reaching their developmental potential by five years of age [2]. Early child development (ECD), especially the period from birth to 3 years, is a period of rapid brain development when children are most susceptible to environmental influences, making it the most critical period of development during the lifespan [3]. In many world regions, the very first indication that children are not thriving is registered at the time of school entry, which is much too late for early intervention.

Clearly, more needs to be done to ensure that very early development, in the first 1000 days [4], is better supported to ensure that optimal developmental trajectories are attainable in these children. Those first years are particularly sensitive because of the intensity of brain development. In the early years of life, the brain is extremely responsive both to positive influences, such as stimulating environment and adequate nurturing care, as well as to negative influences, such as poor nutrition, recurrent infections, lack of responsive and secure parenting, lack of early educational support and unstable economic situations. Many early intervention programs have focused on these factors, and some have demonstrated that giving children a good early start in life can critically influence brain development [3] and thus potentially optimize children’s future economic and social well-being.

One of the obstacles in promoting policies and programs to effectively support children is the lack of validated and reliable measurement tools that could be used to monitor progress for the youngest age groups. While a number of individual and population-level assessments, validated for use in diverse contexts, exist for children at preschool [5], school entry [6] and primary grades [7,8], there is only a small pool of global tools available for children under 3 years. These are mainly developed in Western settings and so costly to implement in low resource settings (e.g., Ages and Stages) or need direct assessment [9], which may require substantial training (Bayley III, Griffiths), making them infeasible for monitoring at the population level. A further challenge is cultural adaptation. If a tool has been specifically developed for a particular country context [10,11], then it may not be readily applied across countries without extensive cultural adaptation, for example, to ensure accurate translation or in the use of relevant props. While some new tools have been created concurrently with the work reported here (e.g., Caregiver Reported Early Developmental Index and Survey of Wellbeing of Young Children) [12], none of them have been developed using an extensive database of child assessments gathered from low- and middle-income countries (LMIC) through a rigorous statistical process [13].

To address these challenges, we recently analyzed 14 datasets comprising 21,083 children; data were collected using seven tools in 10 LMIC countries to identify items that may reliably work across contexts and settings [13]. The resulting prototype tool that we created, the Indicators of Infant and Young Child Development (IYCD), was tested for feasibility of implementation across three sites (Brazil, Malawi and Pakistan) on three continents [14]. At the beginning of this study, the prototype tested comprised 121 items. This tool was created as a simple caregiver-reported tool that was created using tablet-based technology for gathering data (Open Data Kit (ODK)), and it included simple audio/visual media to support understanding of the developmental milestones against which children from 0–3 years were tested.

The present paper outlines the main field testing and final selection of cross-culturally neutral items to validate the WHO IYCD tool across the same three countries, to determine the feasibility of these assessments and to finalize its content. In this study we aimed to determine the performance of the final version of the tool by examining the reliability, ages of attainment and developmental trajectories of each of the IYCD items across countries, as well as in rural and urban settings, in order to ensure that all items in the tool performed consistently across multiple sites, contexts and settings. We also conducted cognitive interviews after to explore language meaning and interpretation in more depth.

## 2. Materials and Methods

### 2.1. Population, Setting and Sampling

#### 2.1.1. Settings and Recruitment

The study was conducted in three countries: Brazil, Malawi and Pakistan. We chose these three countries due to their spread across three continents, the cultural diversity, the feasibility of conducting assessments in both low- and middle-income and rural and urban settings and the availability and interest of the country teams to work with us within a limited time scale. In each country, caregivers of babies and infant children were approached for participation in varying ways depending on the setting.

Brazil. Children and caregivers were recruited by the research team from primary care health centers when parents and children were coming for routine visits or to access any other service such as immunization, pharmacy, dentistry, etc., both in rural and urban areas of Minas Gerais state. The urban sample also included children recruited in a daycare center for children of Sofia Feldman Hospital’s staff. While this daycare can be attended by children of all hospital employees, the cleaners, cooks, administrative staff and technicians constitute the majority of parents.Malawi. Children and caregivers were recruited by Health Surveillance Assistants (community health workers) from urban Blantyre and semi-rural Bangwe regions from primary health care centers when children and their caregivers were attending routine visits (the under-5 clinic).Pakistan. Children and caregivers were recruited through the random sampling of lists of children registered with respective Lady Health Workers.

#### 2.1.2. Informed Consent

Research procedures were approved by local Ethics Committees. Full information on the study was provided in the local language before requesting consent, with frequent back-checking to ensure understanding. Participants were informed of the security procedures used to ensure confidentiality and the organizational provisions for data storage and archiving. Informed consent forms were written to be easily understood by lay persons, enabling them to understand the aims, procedures and potential risks of participation. All information was read aloud, in the presence of an adult witness, to participants who were illiterate. Respondents who decided to participate completed a written consent form, and for participants who were illiterate, a witnessed oral consent and a thumbprint in lieu of a signature was requested. All participants were informed that taking part was voluntary and that they were free to withdraw at any time. It was also made clear that taking part or not taking part would in no way affect care for their child. This consent procedure followed WHO recommendations. Once they agreed to participate, caregivers in each country were given a participant information document and were consented by different cadres of workers who had received in-depth training on the informed consent process through Good Clinical Practice (GCP). This included pediatricians, occupational therapists and physiotherapists (Brazil), nurses (Malawi) and Lady Health Workers (Pakistan). Families were recruited between 1 August 2016 and 1 December 2016.

#### 2.1.3. Sampling

As this was a validation study, it was important that we sampled enough children from across the entire age range in order to adequately test all the items in each age group. In order to achieve adequate representation of children at the higher ages, we sampled children up to 42 months for analysis purposes. A stratified sampling frame was drawn up to quota sample children within eight age group strata (3-month intervals from 0 to 12 months and 6-month intervals from 13 to 42 months) by gender and from urban and rural settings. The sampling grid contained 8 × 2 × 2 = 32 cells with children to be sampled in each cell (*n* = 96 per country) (see Appendix A). To create subsamples for reliability and cognitive interviews, one caregiver of a child from each cell (*n =* 32 per country) was randomly selected and invited to take part in the reliability testing, and three caregivers from different cells in each age stratum per country (*n =* 24 per country) were randomly selected for cognitive interviews.

### 2.2. Exclusion Criteria

The exclusion criteria were informed by existing evidence on health factors that may have affected developmental progress of children in the study. Children who had a mid-upper arm circumference (MUAC) of <12.5 [15] and infants less than six months old with a MUAC of <13 [16] at recruitment were excluded (Figure 1). Children who were unwell on the day of assessment or who had other chronic health needs (including neurodevelopmental disorders, HIV or those with recurrent infections) were excluded from the study prior to recruitment.

### 2.3. Measures

#### 2.3.1. Child Development (IYCD)

The IYCD tool version 1.1 implemented in this study included 121 items: 46 motor (22 fine, 24 gross), 40 language (24 expressive, 16 receptive), and 35 socio-emotional items, which resulted from the feasibility study [14]. The tool was created for use on a tablet-based system using ODK (open data kit) with parent report items. The tool contains items such as: “Does your child reach for and hold objects at least for a few seconds?” (motor) or “Does your child use two words together in a meaningful phrase/speak in short two-word sentence? For example, ‘mama go’, ‘give mama’, ‘daddy gone’.” (language) or “Does your child ever try to imitate your actions around the house?” (socio-emotional). In each country, all items were translated by two local translators who knew the subject and then were back translated by two different language experts with consensus gained at each stage. Back translations in all countries were then checked by the subject matter expert team (M.G., P.K., M.J.) and were streamlined as much as possible between countries.

In the preliminary study, all children were assessed for all items despite their age, enabling us to determine the age at which 50% of children could achieve each item. For this main study, each item of the tool was then placed in order by the age at which approximately 50% of children could achieve it from these preliminary data. A starting point for asking each item was determined using the age of children first achieving the skills based on our previous meta-data synthesis and feasibility work [13,14]. In order to facilitate understanding of the items by caregivers, photos, videos or sounds were provided as appropriate (e.g., a sound example for “Does your child make single sounds like “buh” or “duh” or “muh”?). These were first selected and reviewed to consensus by the authors, then finalized based on feedback from the field teams, who also provided some of the photos and videos. They were presented to the caregivers on tablets for each item that was highlighted in the tool as needing a prompt (Figure 2).

#### 2.3.2. Anthropometry

Anthropometric measurements of every child (weight, height, MUAC and head circumference) were collected as per standard WHO protocols [17]. Children were weighed with an electronic infant scale or an electronic adult weighing scale with reading increments of 10 g. Height or length was taken using an infantometer, recorded to the nearest 1 mm, and head circumference was also measured (using non-stretchable plastic tapes, recorded to the nearest 1 mm). Mid-upper arm circumference was obtained using a standard tape measure to the nearest 1 mm. These measures were used to create a height-for-age *z*-score (HAZ), weight-for-age *z*-score (WAZ) and a weight-for-height *z*-score (WHZ) for each child.

#### 2.3.3. Socioeconomic Status and Family Information

Socioeconomic status (SES) variables included wealth index and maternal education, which were assessed using Demographic & Health Survey (DHS) items standardized for each country [18]. The wealth index comprised 21 items as recommended for multiple assets analysis (water source, fuel use, assets, transportation, animals owned, toilet facilities etc.). Information on the stimulation and learning opportunities offered by the child’s home environment was collected with the family care indicators (FCI) [19] items included in the UNICEF Multiple Indicators Cluster Survey.

### 2.4. Training and Procedures for Data Collection

Training was provided by the local team members/co-authors (Brazil C.R.L.A., Malawi L.M.S., Pakistan M.A.R.), with a consistency check visit by one of the other co-authors (Brazil P.K., Malawi M.G., Pakistan V.C.). Training took place over a period of 2–3 days at each site and was conducted in person at each site. The cost of the training depended on the cost of the trainer visit, hiring a room and transport costs to the training location for children and families attending. The training procedures were monitored so a formal “training the trainer” package could be developed. Training materials are available through the IYCD website (https://ezcollab.who.int/iycd, accessed on 2 June 2021).

Once trained, the country teams conducted assessments with caregivers. Generally, the IYCD assessment was conducted first, followed by the physical measurement of the child. In seven cases in Brazil, three in Malawi and none in Pakistan, the IYCD assessment had to be ceased and reprised on another occasion. The majority of the interviews were recorded using tablets run on Open Data Kit software [20].

#### 2.4.1. Reliability Testing

At the initial assessment, a random subsample (*n =* 96; 32 from each of the three sites) was invited to come back for a second assessment to test consistency of measurement. Inter-rater reliability (two different assessors within the same day at a different time and test-retest reliability (same assessor 3–7 days apart) were conducted in the same setting where the child was seen originally (in Brazil and Malawi, this was done in the health centers, and in Pakistan this was done in homes).

#### 2.4.2. Cognitive Interviews

Nine cognitive interviews in each setting were conducted to establish whether caregivers understood each item. They were also asked how they would explain the question back to the interviewer [21]. Responses were recorded and transcribed verbatim into a spreadsheet for review. All comments were translated into English. Translated responses from cognitive interviews were collated into one document and reviewed by the core team (G.L., G.Mc.C., M.G., M.J., P.K., T.D., V.C.). Input was requested from local investigators (C.R.L.A., L.M.S., M.A.R.) during a teleconference. The results were used to inform decisions to revise wording or retain/delete items as described in the results to ensure optimal clarity of the formulation of all items chosen in the final tool. No formal qualitative analyses were conducted.

### 2.5. Procedures for Finalization of the IYCD

In order to finalize the tool, firstly, each core team member reviewed the evidence of the items’ performance based on 4 criteria: (i) universality (whether the age at which 50% of children passed each item was similar across countries in terms of location) and discrimination (whether the plotted curve trajectories (slopes) adequately demonstrated increasing attainment by age), as described in Lancaster et al. [13], (ii) inter-rater and test-retest reliability, and (iii) responses from the cognitive interviews (verbatim). Secondly, a joint full team meeting was conducted with the core and country teams. This meeting, which included M.G. (neurodevelopmental pediatrician), T.K. (developmental psychologist), M.J (developmental biologist), G.L. (biostatistician), V.C. (pediatric neurologist), T.D. (pediatric neurologist), M.R. (psychologist), C.R.L.A. (pediatrician) and L.S. (psychologist), was convened to reach consensus as to which items were to be included in the final version of the IYCD.

### 2.6. Sample Size

In order to enable item response theory (IRT) methods to be used in the analysis, which was important for building the scoring algorithm [22], we established that a sample of approximately 300 children was required. Considering the time needed to carry out the assessments and collect all the data, as well as available funding and feasibility of assessments in the countries, the sample size was set at 96 children per country, 288 in total. This also allowed us to devise a quota sampling scheme with equal allocation of children to each cell (see Appendix A). For the IRT model used (2PL), 250 children are considered a sufficient sample size [23] and 288 children gives us an 80% power at a two-sided 0.05 alpha to detect as significant a Pearson’s correlation of 0.16 or higher of the tool score against other contextual variables. Thus, we felt the sample size sufficient for the purpose of tool validation.

### 2.7. Statistical Analyses

#### 2.7.1. IYCD Item-by-Item Analyses

An exploratory item-by-item analysis was conducted using logistic regression with the log odds of the probability of passing regressed on the natural logarithm of age to determine how well each item performed across countries. The age at which 50% of children passed an item and the slope of the curve were extracted for each country, and both empirical- and model-based probabilities of passing an item were plotted against age to provide a graphical representation of item trajectories for each country.

#### 2.7.2. Demographic and Contextual Covariates

The demographic profile of the countries was summarized using mean and standard deviation (SD) for continuous data and count (percentage) for categorical data. The age of the child was computed by subtracting the date of birth of the child from the date of administration of the first application of the measure. Decimal age was used throughout, whereby 1 year is equal to 1, 6 months is equal to 0.5 and 3 months is equal to 0.25, and so on. Weight-for-age *z*-scores (WAZ), height-for-age *z*-scores (HAZ) and weight-for-height *z*-scores (WHZ) were constructed according to the WHO 2006 standards [24,25]. They were computed in R using code provided from WHO Anthrostat [26]. In order to make the FCI score comparable across different ages, a generalized partial credit model (GPCM) was used in the R package MIRT [27]. Maternal education was divided into four main categories: no school, primary only, secondary only, and above secondary. In order to make the responses between the DHS wealth index data comparable across countries, we also constructed a two-parameter logistic IRT model using MIRT [27] to create a socioeconomic status (SES) score. The generation of the FCI and SES scores are detailed elsewhere [28].

#### 2.7.3. Missing Data

Missing IYCD item responses (1.6%) were not imputed as the IRT model uses full maximum likelihood estimation. Missing covariate data were imputed using the R package MICE [29]. The numbers of missing data points for covariates were: HAZ *n* = 29; WAZ *n* = 23; maternal education *n* = 5; sex *n* = 1; FCI *n* = 6; SES *n* = 6; urban/rural *n* = 0.

#### 2.7.4. Reliability Analyses

Inter- and intra-rater reliability were calculated for each item using a raw agreement proportion (the proportion of pairs of ratings that agree exactly), Cohen’s kappa statistic as well as Gwet’s AC1 [30], which provides more stable estimates [31] of agreement in situations where there is high trait (pass/fail) prevalence.

#### 2.7.5. Creation of a Development-for-Age *z*-Score (DAZ)

Once the final items had been selected, a scoring system for the final tool was set up. A generalized partial credit model (GPCM) [32] using an empirical histogram prior [33,34] to account for the non-normality in the ability (development) distribution was fitted to the data using the R [35] package MIRT [27]. As the data contained a mixture of binary and three ordinal category responses, a polytomous IRT model was required. Taking the latent scores for each child from the IRT model, the LMS (lambda, mu, sigma) [36,37] method of centile estimation was used to remove the effect of age to create age contingent *z*-scores, which we termed “development-for-age *z*-scores”, or DAZ for short.

#### 2.7.6. Validation

The DAZ scores were plotted against the demographic and contextual variables to explore known-groups construct validity of the tool by comparing countries, maternal education categories and gender. Differences in mean scores were tested using a *t*-test or analysis of variance (ANOVA). Concurrent validity with respect to FCI scores, SES scores, WAZ, HAZ and WHZ was examined using Pearson or Spearman correlation coefficients. A more detailed analysis is described elsewhere [28].

#### 2.7.7. Patient and Public Involvement

We did not directly include patient and public involvement in this study, but all community activities were discussed with health clinic staff and leaders in study areas prior to starting the study. The study team leads received feedback from the local assessors on a regular basis about any problems or difficulties in recruitment within the communities. No issues were reported, and there were no major issues with recruitment.

## 3. Results

Complete data were available for 269 children (Table 1): 97 in Brazil, 77 in Malawi (due to issues with the tablets not functioning appropriately and not saving data) and 95 in Pakistan (Figure 1). Recruitment was generally high in all locations, was always during working hours and was not problematic. Most caregivers consented to their children being included in the study; however, 5 of the 105 caregivers consented but were unable to be assessed in Brazil due to lack of time. In Pakistan, there were no refusals during recruitment. The Pakistan team in particular found recruitment easy as data collectors were from the same community, they spoke the same language and they already had a strong relationship with the community. Similarly, the staff from Malawi College of Medicine worked closely with the health surveillance teams in the local health centers, where recruitment for studies is common and with most parents willing to take part. Only eight (2.7 %) of all children approached did not meet stringent criteria of having a MUAC greater than 12.5 cm (two from Pakistan and six from Malawi). Seventeen children in Malawi and three from Brazil had missing data on developmental milestones, mainly due to issues with the tablets not functioning, missing birth date data and problems uploading data properly. There were low proportions of missing data (0–2%) for maternal education (5), sex (1), urban/rural (6), FCI score (6) and SES (6) and slightly higher numbers for HAZ (29), WAZ (23) and WHZ (29) (9–11%). Missing data were imputed for all covariates.

### 3.1. Sample

Sample characteristics are shown in Table 2. In our sample, almost all (98%) Brazilian mothers had completed secondary education or above and generally had a higher SES index (0.59) and FCI score (0.25) than in other settings. The Brazilian children had HAZ (−0.28) and WAZ (0.05) closest to the WHO standards. In the Malawi sample, very few caregivers went beyond secondary education, with most evenly split between those who completed primary or secondary education; the mean children’s HAZ and WAZ were somewhat below the WHO normal standards (−0.85 and −0.06 respectively). Our Pakistan sample was the least educated, with only a third (36.8%) of mothers completing secondary school education or above. It also had the highest number of children with a low mean HAZ and WAZ (mean −1.1 and −0.92 respectively).

### 3.2. Item Performance

Figure 3 illustrates examples of different item response trajectories.

Plots for all items tested are available in the second Appendix A. Approximately 90% of items fell into the patterns demonstrated in Figure 3a,b, with well-marked developmental progression in item attainment among children within and between each country. Two examples of poor items are illustrated by lines that are very flat (Figure 3e) or that show too much variation in progression between countries (Figure 3f).

A number of items (11/121) did not show a clear developmental trajectory. Nine of eleven poorly performing items belonged to the socio-emotional domain (representing 9/35 or 26% of the items in that domain), and one each belonged to the gross motor and expressive language domains (representing 1/24 or 4% of items in each domain). Items that showed considerable differences in terms of attainment across countries and the poorly performing items were subjected to expert review by the core and country teams. During the review (reported below), it was ascertained whether the item was likely to be exhibiting bias (due for example to misunderstanding or poor translation), or true differences between countries, and therefore whether it should be retained or deleted from the finalized tool.

### 3.3. Reliability

Inter-rater and test-retest reliability statistics were calculated for all 121 items tested. The results are shown for the 90 well-performing items that were eventually retained in the finalized tool (Table 3) and the 33 removed items. We report statistics as a whole across all countries, as numbers for individual countries were too small due to sample sizes. The raw proportion of agreement (RAP) is reported as well as frequencies of items with statistics less than 0.6 (poor/fair/moderate reliability), between 0.61 and 0.80 (good) and above 0.81 (very good) for Cohen’s kappa and Gwet’s AC1 [31]. As it can be seen from the table, the majority of items showed very good reliability, with poorest outcomes in the socio-emotional domain and highest reliability values in the motor domain. Inter-rater reliabilities by domain for the retained items ranged from 0.78 to 0.95. Intra-rater (or test-retest) demonstrated excellent reliabilities, with a range of mean reliability by domain from 0.84 to 0.96.

### 3.4. Cognitive Interviews

Twenty-seven caregivers (nine in each country) were interviewed and in all cases, generally demonstrated a good understanding of the questions that were asked of them. The feedback they provided was used to revise wording of a few items. The four items that showed the highest degree of misunderstanding were: “Does your child walk backwards, two or more steps without any support?” (GRO20), “When you say ‘no’, does your child stop what they are doing?” (REC7), “Can your child complete a five piece puzzle?” (EXP23) and “Can your child understand on first try what is being said to him/her?” (SE12). All these items were removed as shown in Table 4 with item numbers as in Prototype 1 [14].

### 3.5. Finalisation of the Tool

The tool was reviewed and finalized according to the four criteria described in the methods. After the review of items’ performance, including the feedback from cognitive interviews, 100 hundred items were retained, 23 items were marked for deletion (Table 4) and 10 were retained as important non-scoring items (see below). Four of the twenty-three items were revealed to have extensive overlap in the range of age of attainment, and two were therefore deleted. The main reasons for item deletion were, therefore: poor developmental trajectories, poor reliability, wide differences in age attainment between countries and issues with clarity of meaning from cognitive testing. These are listed together with item wording in Table 4. Two additional items were added for the higher ages to fill gaps: LAN19 “Does your child say at least six words?” and LAN22 “Does your child identify at least seven objects?” Agreement was reached by the team to finalize the IYCD. The final list of items is in the Appendix A and has 100 items with 90 of them in 3 domains of development (40 motor items, 30 language and cognitive items and 20 socio-emotional items). Ten behavioral items that showed no or very poor developmental progression on the socio-emotional scale were also retained as important but were not to be scored as part of the IYCD final score.

### 3.6. Validation of IYCD Score Using the DAZ Scoring System

Graphs depicting the 10th, 25th, 50th, 75th and 90th centiles of attainment of each item within a domain were constructed (Figure 4).

Figure 5 shows a density plot of the DAZ scores across countries.

The plots show clear differences between countries in DAZ scores, with Brazil achieving slightly higher developmental scores overall than Pakistan and Malawi. Malawi in turn is slightly ahead of Pakistan in terms of the children in our samples. Mean (SD) DAZ scores by country were Brazil = 0.52 (0.86), Malawi = −0.13 (0.91), Pakistan = −0.43 (0.98), and these were found to be statistically significantly different (*F*(2,266) = 26.87, *p* < 0.001). Table 5 shows that DAZ scores were highly correlated with HAZ, WAZ, maternal education, SES and family environment (FCI). Some example density plots are shown in Figure 5b–d. Figure 5b shows the relationship between DAZ scores by WAZ tercile, which was significant (F(2,243) = 5.657, *p* = 0.004), mean (SD) DAZ score for lower tercile = −0.16 (1.10), middle tercile = −0.08 (0.86), and upper tercile = 0.31 (0.91). Figure 5c shows the significant relationship between SES tercile by DAZ scores (F(2, 260) = 27.00, *p* < 0.001), lower tercile = −0.38 (0.88), middle tercile = −0.21 (1.03), upper tercile = 0.56 (0.77). However, as expected the relationship between sex and DAZ was not significant (*t*(266) = −0.66, *p* = 0.51), mean (SD) DAZ score for males = −0.05 (0.98) and females = 0.03 (1.01) (Figure 5d).

The final version of the tool (IYCD version 1.2) is shown in Appendix A. The tool and all related training materials are available on the WHO IYCD website (https://ezcollab.who.int/iycd, accessed on 2 June 2021).

## 4. Discussion

The WHO 0–3 indicators of Infant and Young Child Development (IYCD) tool is a parent report tool with 100 items (40 fine motor and gross motor, 30 language, 20 socio-emotional and 10 non-scored behavior items) that shows very good reliability, clear developmental trajectories and minimal variability across countries. Through this process, we have established a set of items for children under 3 years of age that can reliably monitor children’s achievement of developmental milestones at approximately the same age across countries. We have further demonstrated that these items can be reported reliably by parents and that the items reflect well the development of children in three countries on three different continents.

Our analyses showed clear developmental trajectories comparable across the three different sites for all items within the motor (gross and fine) and language domains of development. There were fewer items that followed clear developmental trajectories in the socio-emotional development domain. Moreover, some items, though deemed important for healthy development, did not follow the age-based developmental trajectory in the same way as items in the motor and language domain. In response to this, the socio-emotional domain in the IYCD was limited to 20 items that met the developmental criteria. Ten remaining items, due to their relevance and associations with other critical variables, were retained in a separate scale whose scoring is not included in the development-for-age (DAZ) final score.

Recent developments in the field have demonstrated similar results to our own. For example, the results of the study on the Guide for Monitoring Child Development—a more detailed practitioner interview with a parent of a child under 3 years—demonstrated clear and comparable developmental trajectories on milestones across four countries [38] Moreover, research evidence from another new tool created for the same purpose as IYCD, the Caregiver Reported Early Development Instrument (CREDI), has also demonstrated developmental progression of young children’s skills across countries [12] and strong association with family socioeconomic status [39]. This work was conducted concurrently to IYCD, and the teams are now harmonizing for future stages of work [40]. Even though the development of Ages and Stages Questionnaire has been largely based on US-based samples, and it combines observation with parent report, thus not directly comparable with our methodology nor as feasible for large-scale monitoring, results of cross-cultural use support our findings [41]. While all the tools mentioned above have specific advantages, to our best knowledge, none of them used a theoretical framework combined with such an extensive empirical database in the way that the IYCD has.

Despite the diversity of our sample, which came from three countries from three continents with diverse language, culture and customs, our analyses revealed item performance remarkably similar across the sites. We believe that the detailed operating procedures developed prior to the study and closely adhered to throughout supported this outcome. For example, the process of exact translation and back translation from the English to each local language to retain the meaning of items as closely as possible across sites, supported by the local investigators and at least two other team members for each country, ensured high quality control. In following the standard guidelines, we also used exactly the same very specific operating procedures across all sites in assessment of developmental milestones, anthropometry, socioeconomic status and family care. All difficulties and country-specific adjustments were discussed to consensus to ensure that any amendments were within the accepted margins. Our study demonstrated that it is possible to maintain such processes in the measurement of very early child development and that it leads to results comparable across countries. It is our intent to make the procedures available so that they can be utilized on a wider scale in the future.

Despite these strengths, our study had some important limitations. The sample for this study was limited to 269 participants. The next stage in this work would be to conduct a full large-scale validation study across numerous countries and settings to enable us to ensure validity and normal reference values across settings. With the recent nurturing care framework and strategic development goals promoting the need for measurement of early child development as an outcome for programmatic and population level studies [42,43], it is important that efforts to establish valid and comparable developmental monitoring tools move to the next stage. In view of the recently increased interest and volume of evidence on development of children in the first 3 years, the next stages in this work will be to compare and harmonize tools to create one version of a tool that can be upscaled, validated and standardized across multiple countries in order to provide clarity for the international community. This next step is presently underway through the harmonization of work conducted by the CREDI, IYCD and D-Score teams [40]. Our detailed study processes described here in this paper and elsewhere will support these next stages of work ([13,44] Lancaster et al., 2020).

Even though 93% of the original recruited sample was included in the final analysis, there were also data missing due to some inaccuracy in birthdate data collection. As this is so crucial to the creation of developmental trajectories, future studies should attempt to avoid this difficulty, for example, by checking the age and birthdate on several occasions during data collection as an added check to ensure accuracy. Another method to ensure accuracy of age, corrected gestational age and birthdate would be to utilize large birth cohorts for the next stage of work in conducting a wider validation of a tool for global use. We aim to use these types of cohorts for our next stages of work [40]. For this study, we used a paper version to collect data, only using the electronic sources for item demonstrations (e.g., audio sounds or photos or videos illustrating items). In the process of creation of the IYCD, we developed a tablet version that is available for use and can be obtained through the IYCD research team.

## 5. Conclusions

The IYCD has shown excellent reliability and validity to be used in population level measurement of early child development in low-income settings. Further testing is required to ascertain its ability to detect the effects of intervention. Through the detailed process of its development, we created a blueprint for a global adaptation of parent-reported measure of development of children under 3 years of age.

## Figures and Tables

**Figure 1 ijerph-18-06117-f001:**
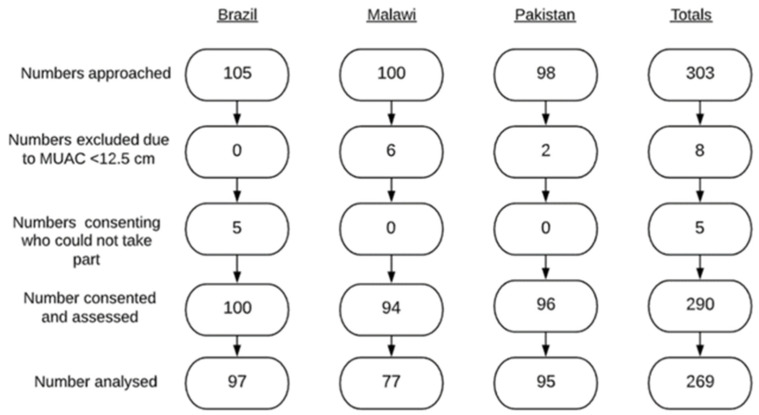
Flow chart demonstrating recruitment to IYCD validation study.

**Figure 2 ijerph-18-06117-f002:**
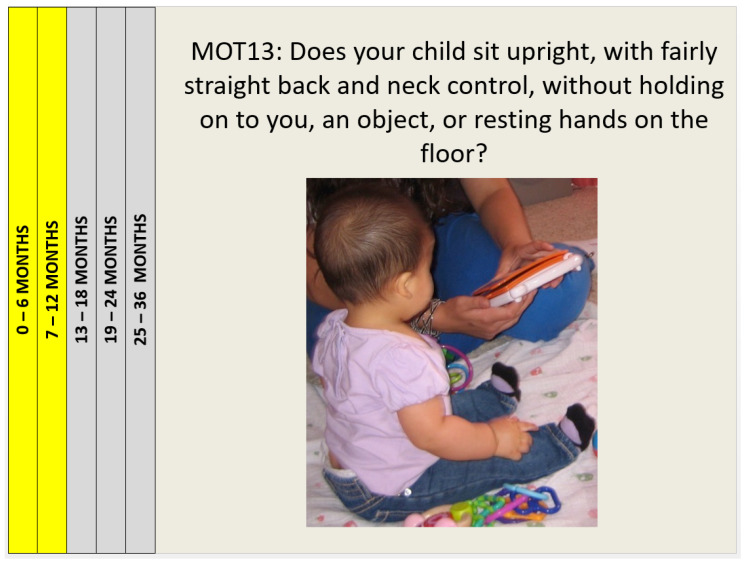
Example of items with prompts.

**Figure 3 ijerph-18-06117-f003:**
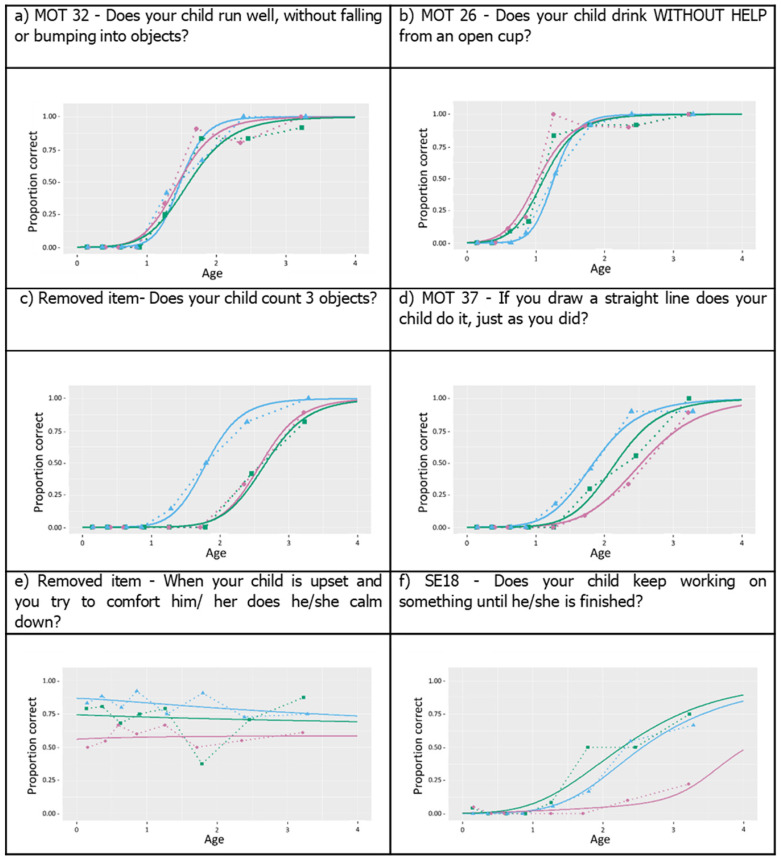
Examples of logistic regression of items. The first two plots, (**a**,**b**), show clear developmental trajectories by age for each country, with agreement between countries. The next two plots, shown in (**c**,**d**), display items that have good developmental trajectories but also some differences between countries. Examples of two poorly performing items are shown in (**e**,**f**). Green—Pakistan, Blue—Malawi, Pink—Brazil.

**Figure 4 ijerph-18-06117-f004:**
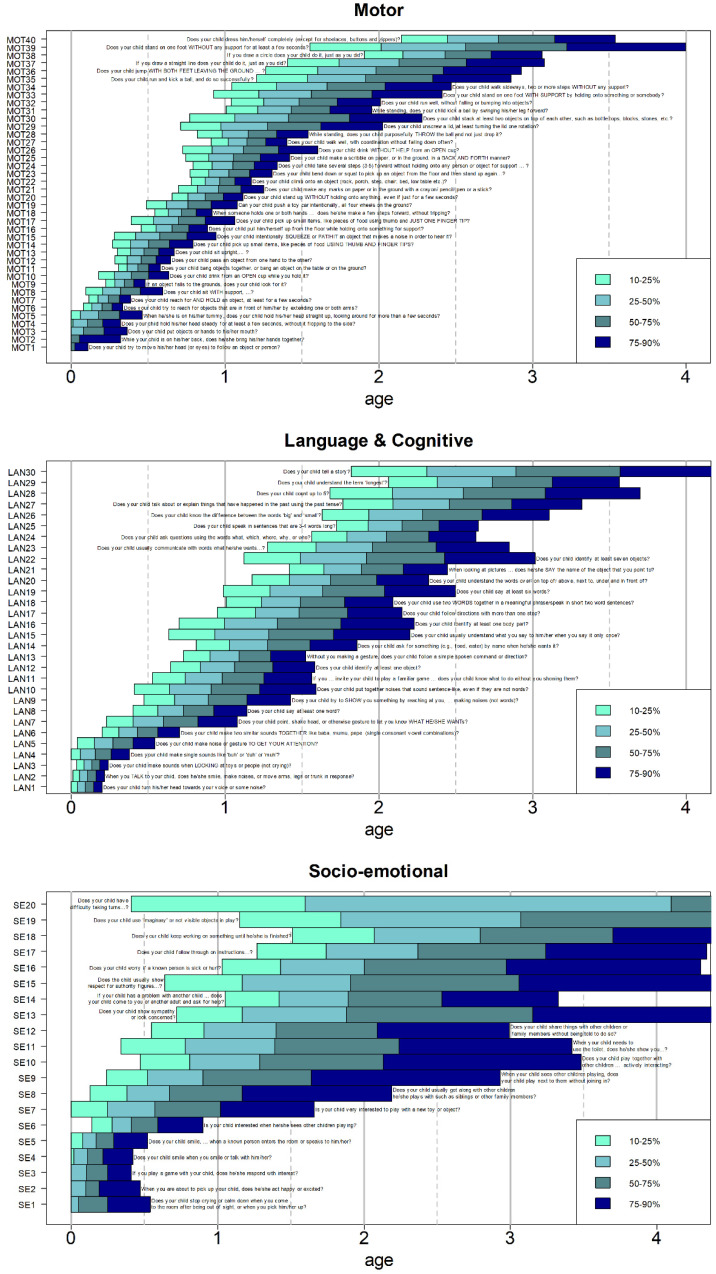
Domains of development with age succession of items with bars showing 10th, 25th, 50th, 75th and 90th centile for attainment across countries.

**Figure 5 ijerph-18-06117-f005:**
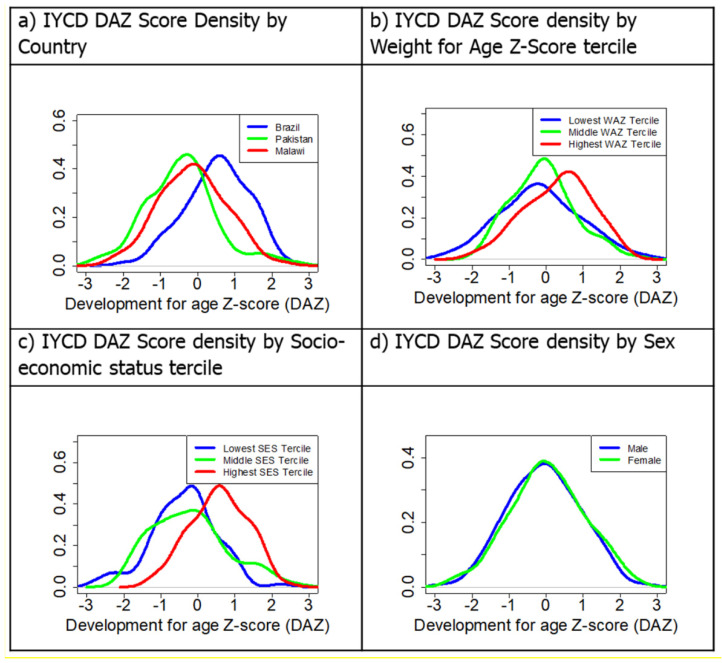
Density plots of development-for-age *z*-scores (DAZ) by country, WAZ, SES and Sex. (**a**) Development for age *z* score by country, (**b**) development for age *z* score varying by weight for age *z* scores broken into terciles, (**c**) development for age *z* score by socioeconomic status broken up into terciles and (**d**) development for age *z* score by sex.

**Table 1 ijerph-18-06117-t001:** Table of children recruited and included in the validation of the Indicators of Young Child Development (IYCD).

Country		Brazil				Malawi				Pakistan				Total
Setting of Those Who Had Data Analyzed		Rural		Urban		Rural		Urban		Rural		Urban		
Sex of Participants		M	F	M	F	M	F	M	F	M	F	M	F	
Age band of participants (months)	0–2	3	2	2	5	2	2	2	2	3	3	3	3	32
	3–6	3	3	3	4	3	2	4	2	3	4	3	3	38
	6–9	3	3	3	1	2	2	3	3	3	3	2	3	31
	9–12	3	3	4	3	2	4	2	2	3	3	3	2	34
	12–18	4	4	2	4	3	0	3	3	3	3	3	3	35
	18–24	3	3	3	3	2	5	2	2	3	2	3	3	34
	24–36	3	3	3	2	2	2	3	2	3	3	3	3	32
	36–42	2	4	3	3	4	3	1	1	3	4	3	3	33
	Total	97				77				95				269

**Table 2 ijerph-18-06117-t002:** Demographic features of children in IYCD sample across countries.

Country	Brazil *n* = 97	Malawi *n* = 77	Pakistan *n* = 95	Total *n* = 269
Demographics				
Age	1.33 (1.01)	1.30 (0.97)	1.34 (1.03)	1.33 (1.00)
M/F	47/50	40/37	47/48	134/135
Anthropometry				
Mean HAZ *** (SD)	−0.28 (1.11)	−0.85 (1.17)	−1.08 (1.0)	−0.72 (1.14)
Mean WAZ **** (SD)	0.05 (1.14)	−0.06 (1.17)	−0.92 (1.04)	−0.31 (1.22)
Maternal education (% total per country)				
No School	0	3 (3.9)	21 (22.1)	24 (8.9)
Primary (%)	2 (2.1)	34 (44.1)	39 (41.1)	75 (27.9)
Secondary (%)	70 (72.2)	36 (46.8)	18 (18.9)	124 (46.1)
Above (%)	25 (25.8)	4 (5.2)	17 (17.9)	46 (17.1)
SES *				
Mean (SD)	0.59 (0.59)	−0.12 (0.59)	−0.47 (0.51)	−0.30 (0.92)
MCS FCI **				
Mean (SD)	0.25 (0.62)	−0.14 (0.72)	−0.15 (0.37)	0.0 (0.61)

* Socioeconomic status index score created from DHS using a two-parameter logistic IRT model. ** Family care indicator index score created from Family Care Indicators using a two-parameter logistic IRT model and GAMLSS for age correction. *** HAZ—height for age *z* score. **** WAZ—weight for age *z* score

**Table 3 ijerph-18-06117-t003:** Reliability frequencies by domain for WHO IYCD items for the 90 retained items and the 33 removed items.

Domain of Development	Type of Reliability Assessed	Mean RAP ***	Kappa <0.6	Kappa >0.6 & <0.8	Kappa >0.8	AC1 <0.6	AC1 >0.6 & <0.8	AC1 >0.8	Total Number of Items
Number of items retained meeting the criteria									
Motor	Inter	0.95	3	8	29	0	1	39	40
	Intra	0.96	0	6	34	0	0	40	
Language and Cognitive	Inter	0.89	5	7	18	2	8	20	30
	Intra	0.94	0	7	23	0	0	30	
Socio-emotional	Inter	0.78	11	8	1	6	9	5	20
	Intra	0.84	5	12	3	0	10	10	
Total items									90
Removed Items **									
Motor	Inter	0.96	1	2	3	0	0	6	6
	Intra	0.96	2	0	4	0	0	6	
Language and Cognitive	Inter	0.88	2	5	2	1	2	6	9
	Intra	0.94	2	2	5	0	1	8	
Socio-emotional	Inter	0.71	17	1	0	10	7	1	18
	Intra	0.79	6	12	0	4	9	5	
Total removed									33

** NOTE: Of the items removed, 9 had inter-rater reliability kappa statistics < 0.40, and 6 had intra-rater reliability kappa statistics < 0.40. These include the 10 behavior items showing no developmental trajectories that were later added to the final tool as important non-scoring items. RAP ***—Raw Agreement Proportion, Kappa—Kappa statistic of agreement, AC1—Gwet’s AC1 agreement statistic.

**Table 4 ijerph-18-06117-t004:** Table of items in IYCD prototype removed and reasons why.

Domain	Item Number from Prototype 1 *	Item Wording	Reason for Removal
Motor	FIN1	Does your child look at your face with interest and attention?	Same age attainment as adjoining items and slightly confusing when translating and back translating.
	FIN24A	Does your child write the first letter of his/her name?	(This item was added in phase I/II.) Item age attainment too advanced for 0- to 36-month children.
	GRO2	When you hold your child in a sitting position, does he/she hold his head steady?	Same age attainment as adjoining items, therefore no need for this item
	GRO3	When pulling your child from lying down on his or her back to sitting, does your child hold his/her head steady?	Same age attainment as adjoining items, therefore no need for this item.
	GRO20N	Does your child walk backwards, two or more steps WITHOUT any support?	Same age attainment to adjoining item; “stands on one foot with support”, which was more understandable on cognitive testing. Reliability not high.
	GRO23	While standing, does your child CATCH a ball and hold on to it, for at least a few seconds?	Same age attainment to GRO21 (Does your child stand on one foot without any support for at least a few seconds?), which requires fewer props.
Language	REC1	Does your child respond or startle when a loud sound is made?	Not related to age (poor developmental trajectory).
	REC2	Does your child respond to your voice or someone else’s voice even if you are not talking to the child directly?	Same age attainment to REC3 (Does your child turn his/her head toward your voice or some noise?) but less consistent across different countries
	REC7	When you say “no”, does your child stop what they are doing?	Same age attainment as adjoining items. Confusing item; not easy to ask on cognitive testing.
	REC18N	If you ask your child “Where is the boy/girl/baby/cow/chicken/etc.?” can your child POINT TO or look at the right picture?)	Same age attainment as REC10 (When you ask “where is the ball/ spoon/ cup/ cloth/ door/ plate/ bucket etc.” does your child look at or point to (or even name) the object? How many objects can your child identify?) but more variability between countries. Very variable direct vs. parent report
	EXP17	When looking at pictures or watching others, can your child tell you what ACTION is taking place (for example running, playing, sleeping etc.)	Same age attainment as adjoining items, and some countries not happy that this item was culturally acceptable.
	EXP18	How many objects can your child name?	Same age attainment as adjoining items, therefore no need for this item.
	EXP22	Can your child explain correctly what the following are used for? Cup (eating/drinking), spoon (eating), knife (cutting), matches (lighting fire, burning things), torch (light), broom	Same age attainment as adjoining items and some cultural differences across countries, making the item less consistent
	EXP25A	Does your child count three objects?	Same age attainment as adjoining items, therefore no need for this item.
	EXP28	Does your child know a song or rhyme from memory?	Attained at a later age, and confusing item for assessors
Socio-emotional	SE3	When your child is upset and you try to comfort him/ her does he/she calm down?	Not related to age (poor developmental trajectory).
	SE7	When you leave your child with a family member, does your child go with that person easily?	Not related to age (poor developmental trajectory).
	SE14	Does child pretend to drink from a cup, or eat with a spoon?	Not related to age (poor developmental trajectory).
	SE16	Does your child care for a doll or stuffed animal as if it were a person (for example by feeding and bathing it)?	Not related to age (poor developmental trajectory).
	SE24	Is your child easily distracted, that is has trouble sticking to any activity?	Not related to age (poor developmental trajectory).
	SE29	When child is very upset, can he/she calm self quickly?	Not related to age (poor developmental trajectory).
	SE30	Would you say that your child bullies or is mean to others at times?	Not related to age (poor developmental trajectory).
	SE32	When you send your child to get something, does he/she forget what he/she was supposed to get?	Not related to age (poor developmental trajectory).

* Please note the item numbers shown here are the item codes from the previous version of the tool.

**Table 5 ijerph-18-06117-t005:** Correlation matrix of main variables.

	DAZ	AGE	SEX	HAZ	WAZ	MAT_ED	SES	FCI
**DAZ**								
**AGE**	0.01							
**SEX**	0.04	0.00						
**HAZ**	0.25 ***	−0.04	0.13					
**WAZ**	0.25 ***	−0.06	0.24 ***	0.56 ***				
**MAT_ED**	0.37 ***	0.10	−0.01	0.29 ***	0.27 ***			
**SES**	0.36 ***	0.04	0.04	0.25 ***	0.10	0.56 ***		
**FCI**	0.22 ***	0.74 ***	0.04	0.00	0.00	0.25 ***	0.24 ***	
**URB/RUR**	0.15 *	−0.05	−0.02	−0.03	0.00	0.20 **	0.22 ***	0.05

Correlation involving an ordinal variable uses Spearman’s correlation coefficient; all others use Pearson’s. SEX: Male = 1, SEX: Female = 2; URB/RUR: rural = 1, URB/RUR: urban = 2. * *p* < 0.05, ** *p* < 0.01, *** *p* < 0.001. DAZ—development-for-age *z* score, HAZ—height for age *z* score, WAZ—weight for age *z* score, MAT_ED—maternal education, SES—socioeconomic status, FCI—family care indicators.

## Data Availability

The dataset supporting the conclusions of this article is available upon request from the WHO team and can be requested through Vanessa Cavallera; cavallerav@who.int.

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
