# Peer review of "Validation of the Infant and Young Child Development (IYCD) Indicators in Three Countries: Brazil, Malawi and Pakistan"

_ijerph, 2021, doi:10.3390/ijerph18116117_

Round 1

Reviewer 1 Report

To validate a population-based measurement tool - Infant and Young Child Development (IYCD), this paper assessed attainment of 121 IYCD items in 269 children aged 0-3 from Pakistan, Malawi and Brazil. Consensus choice of final items depended on developmental trajectory, age of attainment, invariance, reliability and acceptability between countries. Paper found the IYCD Items were acceptable, performed well in cognitive testing, had similar developmental trajectories and high reliability across countries. Development for Age (DAZ) scores showed very good known-groups validity. This is important to validate other ECD measuring tools for sure.

  1. My main concern of this paper is the motivation and discussion, as I know, there is a lot of good tools to measure ECD, like CREDI (McCoy, D. C., Waldman, M., Team, C. F., & Fink, G. (2018). Measuring early childhood development at a global scale: evidence from the Caregiver-Reported early development instruments. Early childhood research quarterly, 45, 58-68.), ASQ, ASQ-SE, of course we need to mention Bayley. There is more, but I don’t see the motivation about these measures and discussions

  1. Related to comment 1, I it would be good if they can compare their measures with these dominant ones – ASQ, CREDI and Bayley…

  1. In method section, it is not very clear to me in terms of sampling, to me it is hard to say “population based”. It would be great if you can provide more information about this.

  1. About missing data, please report it explicitly, what was missed, how many?

  1. I still want to mention the sample size – doesn’t seem to the sample is big enough

  1. There is a typo in abstract - behaviour.

Reviewer 2 Report

This is a well written and important paper that will be of great interest to policy makers and researchers. It provides a thorough description of the results and the comments provided below are minor but necessary for providing a robust justification for the use of the IYCD and the decisions made by researchers and the claims made in the discussion. Best wishes in the next iteration of the paper.

Abstract: Recommend in second sentence that it reads “…too provide reliable estimates of. young children’s developmental status.”

Introduction, paragraph 2: I would recommend neuroscience research (Shonkoff for example) is drawn on here to substantiate your point about the early years of life. I would also recommend that the authors provide information about the features of the WHO IYCD tool and why it was chosen for this study which is key background content for the reader. I would also recommend that the authors provide a justification for the setting – eg. Countries, sites that were chosen in the study and the exclusion and inclusion of items used in the study. Why were these decisions made and what informed them? Furthermore, cultural adaptation is a key point raised by researcher as a challenge – how did these decisions address these challenges? These are presented to readers as facts and would be stronger if they included the research-informed decisions that was used by the researchers prior to the implementation of the research and will be an important framework for the final discussion.

I would also recommend in the discussion section that the evidence from the results is used to substantiate the claims made in the paper – for example, can reliably monitor children’s achievement ….across countries, can be reported reliably, item performance remarkable similar across sites. In addition further discussion around the limitations of the study or next steps could be drawn out further to strengthen your claims. These would be of value in the discussion rather than as a statement raised in the conclusion.

Reviewer 3 Report

Overall

Thank you for the opportunity to review this manuscript.

This study addresses an important gap in the global need to monitor early childhood development. It assessed the attainment of items from the IYCD in young children from three countries to reach a consensus of the final items for this developmental measure. This is with the aim of achieving a global population based measurement of developmental status.

Overall, this paper is well written. It is well-organised and clear and the authors have addressed the objective. There are some sections that would benefit from further development. While reference is made to the development of the IYCD, the background could include some additional information on previous work; and the methods would benefit from some additional detail.  It would be helpful to include the aims in the abstract and refine the conclusion.

Specific comments

2.2 Exclusion criteria

It would be helpful for the reader if there was additional information on the reasoning for the exclusion criteria.

2.3 Measures

This section would benefit from some further development.

2.3.1 Some further detail, without having to go to the cited article would be helpful. While the references will have this detail, it is essential for this paper, in view of the focus being on the suitability of this tool for a global audience,  “Items on the tool were placed in order by the age at which approximately 50% of children could achieve it.”  Is this based on the original IYCD development work? Is this 50% of the original sample or this current sample?

There is no detail about how the measure is administered or scored.  It is not described as a ‘parent completed’ tool here. I am interested to know how the practitioner decides on the starting point and how/if a ceiling is reached?  It is not clear how long it takes to administer the tool or that it was translated into the languages of the communities. While the discussion does make mention of the translation/back-translation process, this could be included here.

2.4 Training and procedures

There is reference much later to the availability of the training materials being available on the WHO IYCD website. However, it would be useful to have that referred to in this section.  Some description of the time/cost/feasibility of the training would be relevant to readers.   Mention is made of the recording of the interviews in this section (2.4),  however, perhaps more relevant to include this in section 2.4.2 with information on Cognitive interviews.  This section could benefit from further detail. No mention is made of whether consent for recording was obtained? Who conducted the interview? Was there an interview scheduled? How were the interview participants selected? Who translated the responses? While these data “were used to inform decisions to revise wording or retain/delete items to ensure optimal clarity” there is no explanation of the translation process and how the potential for bias was addressed.

2.7.8 Could a reason for not including consumers in some way be included/addressed?

  1. Results

While encouraging on the one hand that recruitment “was not problematic” I’m not certain that this is always a measure of success.  True informed consent would result in some families saying no.  Is this a vulnerable population?  Are parents clear that they will not be disadvantaged or excluded from receiving health services if they choose not to participate? Some further information about what was included in the participant information document would be useful. 

2.5 Finalisaton of the tool (is this is the wrong section – if follows 3.4 but is numbered 2.5)

Could the roles of the team, who reached consensus on the final version, be described?

  1. Discussion/Conclusion

This section is logically presented and well written.

Line 81 Mention is made of the “detailed feasibility and pilot studies of process”.  Are these future studies? Feasibility is an important aspect of introducing a measure such as this.  Could this be elaborated on? The authors then describe in the Conclusion that the IYCD has shown “excellent…feasibility” (line 93).   The aims did not specify feasibility as I read it. 

Round 2

Reviewer 1 Report

Thanks for the responses!